# Towards Mapping of the Human Brain N-Glycome with Standardized Graphitic Carbon Chromatography

**DOI:** 10.3390/biom12010085

**Published:** 2022-01-06

**Authors:** Johannes Helm, Lena Hirtler, Friedrich Altmann

**Affiliations:** 1Department of Chemistry, Institute of Biochemistry, Universität für Bodenkultur Wien, Muthgasse 11, A-1190 Vienna, Austria; johannes.helm@boku.ac.at; 2Center for Anatomy and Cell Biology, Division of Anatomy, Medical University of Vienna, Währinger Straße 13, A-1090 Vienna, Austria; lena.hirtler@meduniwien.ac.at

**Keywords:** brain, N-glycans, porous graphitic carbon, brain glycomics, negative mode CID

## Abstract

The brain N-glycome is known to be crucial for many biological functions, including its involvement in neuronal diseases. Although large structural studies of brain N-glycans were recently carried out, a comprehensive isomer-specific structural analysis has still not been achieved, as indicated by the recent discovery of novel structures with galactosylated bisecting GlcNAc. Here, we present a detailed, isomer-specific analysis of the human brain N-glycome based on standardized porous graphitic carbon (PGC)-LC-MS/MS. To achieve this goal, we biosynthesized glycans with substitutions typically occurring in the brain N-glycome and acquired their normalized retention times. Comparison of these values with the standardized retention times of neutral and desialylated N-glycan fractions of the human brain led to unambiguous isomer specific assignment of most major peaks. Profound differences in the glycan structures between naturally neutral and desialylated glycans were found. The neutral and sialylated N-glycans derive from diverging biosynthetic pathways and are biosynthetically finished end products, rather than just partially processed intermediates. The focus on structural glycomics defined the structure of human brain N-glycans, amongst these are HNK-1 containing glycans, a bisecting sialyl-lactose and structures with fucose and *N*-acetylgalactosamine on the same arm, the so-called LDNF epitope often associated with parasitic worms.

## 1. Introduction

N-glycosylation, one of the most common post translational modifications, is known to confer crucial biological functions to its acceptors. N-glycan structures are involved in the folding, secretion and intracellular transport of glycoproteins and are increasingly considered as markers of health and disease. Noteworthy, nearly all congenital disorders of glycosylation are characterized by neurological disfunctions [1]. Other recent studies have revealed a clear connection between Alzheimer’s disease and aberrant glycosylation [2,3]. It was shown that especially fucosylated and oligomannosidic structures are dysregulated in regard to Alzheimer´s disease [4]. The importance of brain N-glycosylation was further shown by the finding that sialylated N-glycans can modulate the neurotransmitter release in nerve terminals [5,6]. Because of its obviously important role in the central nervous system, large structural and functional studies regarding the brain N-glycome have been carried out in the last years [7,8].

A high amount of oligomannosidic glycans (Man5-Man9), representing about 15% of the total N-glycan pool, has been identified in the mammalian brain [9]. Oligomannosidic glycans were shown to be involved in the interaction between the neuronal adhesion glycoproteins L1 and N-CAM [10], and therefore play a potentially important role in neural tissue development. Another interesting feature specific to brain N-glycans is the presence of bisecting GlcNAc within the complex- and hybrid-type glycans, often combined with the occurrence of core- and/or outer-arm fucose [9,11]. Bisecting GlcNAc, which is transferred by Gn-TIII (N-acetylglucosaminyltransferase-III), is able to suppress terminal modifications [12] and plays a role in the progression of certain types of cancer [13]. Bisecting GlcNAc is highly expressed in the nervous system, especially in neurons [14], and is involved in Alzheimer´s disease [15]. The exact mechanism of how Gn-TIII acts on target proteins is still unknown, but a very recent study showed that the enzyme recognizes aspects (or features) of the tertiary sequence of the target protein [16]. Lewis X (LeX) epitopes were identified on several neural glycoproteins, for example CD2420 and Synapsin I [17] and brain tissue [18]. The LeX epitope confers regulatory functions during brain development and plays a role in synaptic plasticity, as reviewed recently [19]. A study conducted by Lee et al. investigated the human brain N-glycome in a temporal and spatial context, and found significant differences between regions of the brain and between age groups [8].

As N-glycans are biosynthesized template-free and are a product of stepwise and competing glycosyltransferase- and glycosidase-activities, the arising structural complexity is hard to predict and represents an analytical challenge. This challenge is further exacerbated by the occurrence of multiple structural and linkage isomers. Alterations in the linkages between α2,6 and α2,3 sialic acids or differences in antenna fucosylation are often correlated with cancer and infectious diseases [20,21]. Endogenous sialic acid-binding lectins (“siglecs”) recognize the linkage between the sialic acid and the next sugar residue [22]. These examples indicate the usefulness of an isomer-specific analysis of the human brain N-glycome, which could open the door to many potential therapeutic and diagnostic applications.

Due to methodological limitations, most studies regarding the structural elucidation of the human brain N-glycome refrained from definitely assigning N-glycan structures. Identification of glycans with positive mode MS/MS, as carried out by Lee et al. [8], is also hampered by the preference of certain sugar-residues to rearrange their position during fragmentation [23,24,25]. A very recent publication about a modularization strategy for site-specific N-glycan structure analysis identified 600 N-glycans and 1501 glycosylation sites on 945 glycoproteins in the brain, but also abstained from definitely assigning structures, as the applied method of positive mode collision induced dissociation (posCID) is basically unable to distinguish between arm isomers and is prone to fucose migration [26]. Other studies regarding the brain N-glycome likewise devoted their attention to aspects other than exact structures [7,27]. The recent discovery of novel structures with galactosylated bisecting GlcNAc in the man brain [28] indicated the need for a more detailed structural analysis.

Here, we present a comprehensive, isomer-specific analysis of the human brain N-glycome based on standardized porous graphitic carbon (PGC)-LC-MS/MS. To achieve this goal, we extended the range of synthetic reference structures with N-glycans substituted with typically occurring features in the brain N-glycome and acquired their normalized retention times as described in our previous publication [28]. The neutral and the charged structures were separated by PGC-solid phase extraction and pools were analyzed separately. Prior to analysis, the sialylated pool was enzymatically desialylated to get information about the underlying neutral scaffolds. Comparison of the standardized retention times of neutral and desialylated N-glycan fractions of the human brain with the biosynthesized standards led to an unambiguous assignment of most peaks, including linkage-isomers. Fundamental differences in the glycan-structures between the nonsialylated and desialylated glycans were found at the composition and isomer level. The presented method proved to be a powerful tool for structural glycomics, as substantiated also by the unveiling of six, partially isomeric, HNK-1 containing N-glycans.

## 2. Materials and Methods

### 2.1. Materials

Samples from two human brains originated from voluntary body donations to the Center for Anatomy and Cell Biology of the Medical University of Vienna. The body donors consented prior to their death to the use of their body for teaching and science. Porcine brain was kindly provided by Christian Draxl from the “Österreichische Schweineprüfanstalt”. The biantennary N-glycan A^4^A^4^ (see Appendix A for structure depictions) with ^13^C_2_ acetyl groups was purchased from Asparia glycomics (San Sebastian, Spain). ^13^C_6_-galactose was purchased from Cambridge Isotope Laboratories (Tewksbury, MA, USA).

N-glycans from human brain were obtained from SDS extracts as recently described [28]. Neutral and sialylated N-glycans were separated with a PGC cartridge [29]. N-glycans from porcine brain, bovine fibrin, beans and human IgG were prepared as described recently [30]. A part of the IgG N-glycans was fractionated by semi-preparative HPLC on a Hypercarb column as described previously [30]. The major components GnGnF^6^, A^4^GnF^6^, GnA^4^F^6^ and A^4^A^4^F^6^ were eluted in this order [31] and could be obtained as isolated fractions for the preparation of defined isomers. All glycans were finally reduced with sodium borohydride. Details of the preparations are given in the Appendix A.

### 2.2. Biosynthesis of Isotope-Encoded Standard Glycan Structures

Recombinant human α1,2-fucosyltransferase (Fuc-TII) and recombinant human β1,4-galactosyltransferase (b4Gal-T1) variant Y285L, used to transfer UDP-GalNAc to non-reducing end GlcNAc-residues were purchased from Bio-techne (Minneapolis, MI, USA). Recombinant bovine β1,3-galactosyltransferase (b3Gal-T) was obtained from Chemily Glycoscience (Peachtree Corners, GA, USA) and bovine milk β1,4-galactosyltransferase (b4Gal-T), bovine kidney fucosidase and jack bean α-mannosidase were purchased from Sigma-Aldrich (Vienna, Austria). The β1,4-specific galactosidase from Aspergillus oryzae was prepared as described in [32]. α1,3/4-specific fucosidase and α1,6-specific mannosidase were purchased from NEB (Ipswich, MA, USA). A β1,3-specific galactosidase was obtained from NEB (P0726S; Ipswich, MA, USA).

Soluble His_6_-tagged forms of human fucosyltransferase III (Fuc-TIII) lacking the N-terminal 34 amino acids, human fucosyltransferase IV (Fuc-TIV) lacking the N-terminal 172 amino acid and human β1,3-galactosyltransferase (b3Gal-T5) lacking the N-terminal 29 amino acids were expressed in the baculovirus insect cell system as described previously [33].

The enzymes were purified by metal chelate chromatography. The buffer to 25 mM Tris/HCl pH 7.4 supplemented with 100 mM NaCl and the volume was reduced to less than 1 mL with an Amicon 10 kDa cut-off membrane (Sigma Aldrich, Vienna, Austria). The enzymes were directly used after purification or stored at 4 °C.

UDP-^13^C_6_-galactose was enzymatically prepared from ^13^C_6_-galactose (Cambridge Isotopes, Andover, MA, USA) and purified by PGC chromatography [34].

Isotope-differentiated sets of reference structures were generated from different scaffold glycans by application of the different glycosidases and glycosyltransferases as detailed in Appendix A. After each enzymatic step, glycans were purified using PGC solid phase cartridges (Multi-Sep Hypercarb 25 mg, Thermo Scientific, Vienna) [34]. Completeness of the enzymatic reaction was checked using PGC-LC-ESI-MS.

### 2.3. Mass Spectrometric Analysis

The purified samples were loaded on a PGC column (100 mm × 0.32 mm, 5 µm particle size, Thermo Scientific, Waltham, MA, USA) with 10 mM ammonium bicarbonate as the aqueous solvent A and 80% acetonitrile in solvent A as solvent B, as described in [28]. In brief, 5.5 min after sample application at 1% B, a gradient from 8 to 22% solvent B was developed over 52.5 min followed by an increase up to 68% B at a flow rate of 6 µL/min. Detection was performed with an ion trap mass spectrometer (amaZon speed ETD; Bruker, Bremen, Germany) equipped with the standard ESI source directly linked to the Thermo Ultimate 3000 UPLC system. MS scans were recorded in positive and/or negative mode from 400–1600 *m*/*z*. Standard source settings (capillary voltage 4.5 kV, nebulizer gas pressure 0.5 bar, drying gas 5 L/min, 200 °C) were used. Instrument tuning was optimized for a low mass range (around 1500–2000 Da). MS/MS was carried out in data-dependent acquisition mode (switching to MS/MS mode for eluted peaks). Data interpretation was done with DataAnalysis 4.0 (Bruker, Bremen, Germany).

### 2.4. Binary Gradient Mixed-Mode Chromatography of Fluorescent Derivatives

Analytical separation of differently charged glycans was done by modification of a protocol that used a ternary gradient [35]. A binary gradient was formed from solvent A being 80% acetonitrile in H_2_O and solvent B being 250 mM ammonium formiate. A gradient from 0 to 35% B in 30 min at a flow rate of 0.3 mL/min was applied to an anion exchange column (Phenomenex, Luna 3 µm NH_2_, 150 × 2.0 mm). Aminobenzamide labeled glycans were detected by their fluorescence excited at 330 nm and monitored at 420 nm.

## 3. Results

### 3.1. Biosynthesis of Glycan Standards

Guided by reports that brain N-glycans are highly fucosylated and bisected and contain galactose primarily in β1,4-linkage but also, to a smaller extent, in β1,3-linkage [9,11], we bio-synthesized a range of N-glycans containing these structural features in addition to the almost complete set of permutations of glycans with five hexose, four N-acetylhexosamine and one fucose residue (H5N4F1) generated for a recent work [28]. Some studies also reported α1,2-linked fucose to galactose [36,37] as being present in the brain glycome, so we also biosynthesized structures containing this feature. To consider the possible presence of Lewis A (LeA) determinants, we also synthesized glycans containing this substitution.

A set of structures comprising biantennary glycans with bisecting GlcNAc was prepared with GnGnF^6^bi from pig brain as the starting point (Figure 1). The scaffold glycan was incompletely β1,4-, or β1,3-galactosylated with ^13^C_6_-UDP-galactose, which introduced a mass-increment of 6 or 12 Da. Fuc-TIII and Fuc-TIV were then applied to build LeA- or LeX epitopes, respectively. Finally, a part of the structure was incubated with fucosidase from bovine kidney to remove core-fucose (Figure 1). Assignment of arm isomers was accomplished with—in this case unambiguous—positive mode CID (posCID) as shown in Appendix A [38].

Another set of structures without bisecting GlcNAc was generated in a similar way, starting with the IgG glycan ensemble A^4^A^4^F^6^, A^4^GnF^6^, GnA^4^F^6^ and GnGnF^6^. The scaffolds were converted to the desired standards with the help of b3Gal-T, fungal galactosidase, Fuc-TIII and Fuc-TIV (Appendix A). In contrast to Fuc-TIV, Fuc-TIII is able to generate both LeA- and LeX-epitopes. ^13^C_2_-acetylated A^4^A^4^ was partly digested with fungal galactosidase, incubated with b3Gal-T and ^13^C_6_-UDP-galactose and finally treated with Fuc-TII to create the H5N4F2 structures with blood group H- (bgH) epitopes on both arms (Appendix A). The H4N4F1 standards with a bgH, LeA or LeX-epitope, the H5N4F3 structures with a core-fucose and two α1,3- or two α1,4-fucose residues and the H4N4F2 and H5N4F2 with a LeA-epitope and core fucose were biosynthesized, as shown in the detailed biosynthesis pathways in Appendix A.

Structures containing LacdiNAc-epitopes were biosynthesized in a similar way using either GnGnF^6^bi or the mixed IgG glycans A^4^A^4^F^6^, A^4^GnF^6^ and GnA^4^F^6^ as the starting points. B4Gal-T1-Y285L was used to catalyze the transfer of GalNAc to GlcNAc-residues on the non-reducing end. The resulting structures were either treated with fucosidase from bovine kidney or fungal galactosidase. Detailed biosynthesis pathways for these structures are shown in Appendix A.

The recently generated library of forty-one N-glycans of composition H5N4F1 [28] was augmented with structures having less hexoses and/or more fucose residues (H4N4F1–2 and H5N4F2-3), as detailed in the Appendix A.

### 3.2. The Virtual Minute Retention Time Library

The products of each step were blended with the isotope-labeled time-grid standards (Tigr mix) [28] and subjected to PGC-LC-ESI-MS/MS. The retention times were converted to “virtual minutes” (vimin), as described in our earlier report [28] and fed into the Tigr glycan library (Appendix A). The time grid approach certainly yields a more useful description of a glycan´s retention than a simple one-point relation. Nevertheless, an error margin of up to 0.1 min must be conceded to the vimin values. As expected, the elution order of the standards followed the already known PGC elution rules. Structures containing bisecting GlcNAc eluted earlier than their non-bisected counterparts, α1,6-core fucosylated increased the retention and LeX-fucose decreased the retention. The influence of LeA- and bgH-fucose on the retention is hardly predictable and depends on arm-position and linkage-type of the galactose residue.

### 3.3. Analysis of Brain N-Glycans—The Concept

The biosynthesis of glycan standards formed the basis for embarking on the isomer specific analysis of brain glycans, whereby our primary interest was in complex-type structures. About 60% of brain N-glycans are neutral as judged from mixed-mode fluorescence HPLC, the complementing 40% comprise sialylated glycans and sulfated N-glycans, which account for a notable 4% of the total as judged from HPLC after desialylation (Appendix A). The sulfated fraction comprised various sulfated glycans, notably some with the HNK-1 epitope [39]. To facilitate structural analysis of the sialylated glycan pool, which is not only of low abundance but also diversified by different sialic acid linkages, we at first separated neutral and charged glycans by passage over a PGC cartridge [29] and then removed sialic acids to allow thorough isomer specific analysis of the neutral backbones by the joint application of the virtual PGC retention library, posCID and negCID—and in some cases digestion with bovine kidney fucosidase. Most of the glycans with 3–5 hexoses, 4–6 N-acetylhexosamines and 0–3 fucoses could thus be structurally defined. A few more glycans could be unambiguously assigned despite the lack of biosynthetic standards as detailed below. A comprehensive overview of features of these peaks is provided as Appendix A. Appendix A also lists structures and their features that could not be identified by retention time and CID spectrum (black dots in Figure 2D map). These characteristics nevertheless provide such glycans with a traceable identity, even though their exact structure is not yet known.

### 3.4. Analysis of Brain N-Glycans—Neutral Structures

Many structures could be unambiguously assigned by retention time with reassurance from negCID and posCID (Figure 2 and Appendix A). In the following, structures that deserve additional comments and structures that could be elucidated in the absence of identical reference glycans will be discussed. Peaks will be identified by their number of hexoses, N-acetylhexosamines, fucoses and their standardized retention time.

Peak 351-17.2 (Figure 2A) is a known major constituent of the brain N-glycome [6,9,40]. The non-fucosylated version 350-11.5 was identified as GnGnbi. Peak 351-27.2 exhibited a characteristic 407 *m*/*z* peak in posCID, indicating a GalNAcβ1-4GlcNAc (LacdiNAc) unit. This could be located to the 6 arm with the help of the diagnostic D and D-18 ions in negCID (Appendix A). As this peak is not coeluting with the standard AnGnF^6^ (eluting at 32.2 vimin), it is most likely AnMF^6^bi. The presence of LacdiNAc containing glycans in the human brain was already shown [8], but without explicit structure suggestions. Notably, in the authors’ hands, only b4Gal-T, but not b3Gal-T5 accepted UDP-GalNAc as donor substrate. Hence, the terms AnGnF^6^ and An^4^GnF^6^ have an equivalent meaning, as the linkage to the GalNAc-unit can only be a β1,4-linkage and therefore the linkage information in the abbreviation would be redundant.

No standard was likewise on hand for the 352-29.4 peak with a posCID fragment at *m*/*z* 553, which would point at a LacdiNAc unit plus fucose, and which would identify the glycan as being (AnF)GnF^6^). Aware of the perils of posCID [24,25,28], we took a closer look at structures with the *m*/*z* 553 signature (see separate chapter below).

Compound 440-11.2 could—despite the absence of a standard—be clearly identified as Man4Gnbi, or even more precisely as M^3^Gnbi, due to its early elution time indicative of bisection, the D-18 ion at *m*/*z* = 467 in negCID and the strong prevalence of α1,3-Man in Man4 N-glycans [28,39] (Appendix A). A mechanistic study on α-mannosidase II corroborates these deliberations [41]. The core-fucoslyated analog 441-15.5 could be identified as Man4GnF^6^bi = M^3^GnF^6^bi via fucosidase digestion and the core fucose specific fragment at *m*/*z* = 350 (Appendix A). The minor peak 451-20.5 coeluted with a peak for the LeX standards Gn(AF) and (AF)Gn, which are not separable with the applied chromatographic system. Similarly, the large peak 442-29.6 could constitute either (AF)GnF^6^ or Gn(AF)F^6^. The occurrence of diagnostic D- and D-18-ions at *m*/*z* = 834 and 816 z, respectively (Appendix A), shows the presence of (AF)GnF^6^.

The 451 level was populated with at least nine different peaks, which in part may contain unseparated isomers. An example of this phenomenon is peak 451-12.0, which coeluted with the unseparated standards (AF)Gnbi/Gn(AF)bi. Lack of a D-ion corroborates the observation that bisected structures rather generate D-18 ions [42]. The strong dominance of the D-18 ions at 816 over *m*/*z* = 508 suggested (AF)Gnbi as the dominant structure and an at least essential stability of the D-18 ion (Appendix A).

Composition 541 has been amply studied in the preceding work that revealed the substitution of bisecting GlcNAc in the case of M^3^Gn(AF)-bi and M^3^GnF^6^(AF)-bi [28]. Only two of the peaks occur in both the neutral and acidic fraction. In the case of 541-22.7, this necessarily implies that the galactose on the bisecting GlcNAc had carried the sialic acid in the form of a bisecting Sia-Gal-GlcNAc chain, in other words a bisecting sialyl-lactose, a truly peculiar novel structural motif.

The 551, 552 and 553 levels contained easily identifiable diantennary glycans from A^4^A^4^F^6^bi to (AF)(AF)F^6^bi and a number of triantennary glycans, whose exact structure lies outside the scope of this work.

A range of rather abundant neutral N-glycans showed glycan compositions, which were not covered by our biosynthesized standards. To make the current survey more complete, the retention times of these glycans were normalized and their features are taken up in Appendix A. As far as possible, the structures were characterized with the help of negative mode MS/MS.

The rather small but abundant N-glycans 220-16.5, 221-28.1, 320-26.5 and 321-36.5 were assigned on the basis of IgG glycan degradation products [31] and diagnostic negative mode CID fragments (Appendix A) (Appendix A). The 432-41.8 peak was identified as (AF)MF^6^ with the help of the diagnostic negCID fragments *m*/*z* = 834.3 and 350 (Appendix A), and its only possible source being a LeX containing glycan.

With basically no other possibility, the small peak right after 432-41.8 was assigned as the arm isomer M(AF)F^6^. Bovine fucosidase converted the 432-41.8 peak into a 431-31.7 peak that necessarily contained an outer arm fucose. Thus, the 431-31.7 N-glycan was assigned as (AF)M, while the later eluting 431 peak at about 43.1 vimin (Appendix A) was identified as A^4^MF^6^ with negCID (Appendix A). The D- and D-18 ions of *m*/*z* 485 and 467, the F-ion at *m*/*z* 570 and core fucose specific fragment at *m*/*z* 350 (Appendix A) identified 532-25.2 vimin as the hybrid type glycan Man4(AF)F^6^. By a similar rationale, 341-28.1 was assigned as GnGnF^6^ (Appendix A) and the earlier eluting 341-14.7 peak with a less abundant ^1,3^A_2_-ion as GnMF^6^bi (Appendix A), a structure that was already described to be the product of a brain specific hexaminidase [43].

The isomeric structures of oligomannosidic glycans were assigned, according to previous work (Appendix A) [44].

### 3.5. Brain N-Glycans with an LDNF Determinant

Sensitized by the 352-29.4 peak as containing an LDNF structure, we facilitated negCID of interesting structures by prior HILIC fractionation of glycans from—in this case—the *Lobus frontalis*, and on top of that, simplified the glycan pool by removal of the core-fucose. An *m*/*z* = 553 fragment thus could hardly be caused by fucose migration [25,28]. In fact, two 361 N-glycans—arising from 362-17.7 and 362-32.4 after bovine kidney fucosidase digestion—yielded D-221 or D-18 ions of *m*/*z* = 857.3 in negCID, proving the presence of LDNF units (Figure 3). The original 362 structures (AnF)GnF^6^bi, (AnF)AnF^6^ and their respective arm isomers exhibited 3% and 5% peak height, respectively, as compared to GnGnF^6^bi.

### 3.6. Analysis of Brain N-Glycans—De-Sialylated Structures

Visual comparison of the mass spectra of the non-sialylated and the desialylated N-glycan pool already revealed fundamental differences between those fractions. A few peaks were found in both fractions, but most of the desialylated fraction peaks occurred just there. In the following, these previously sialylated peaks shall be discussed.

The single 351 peak at 32.5 vimin (Figure 4A) contained GalNAc and core-fucose (Appendix A) and coeluted with the standard AnGnF^6^. Obviously, sialic acid was linked to GalNAc in this structure. This rarely reported element was previously observed in brain sections [8] and originally in mammalian non-brain proteins [45,46,47,48].

The intense peaks 441-32.9 and 441-33.7 were identified as A^4^GnF^6^ and GnA^4^F^6^, respectively, albeit in a reverse ratio to that seen in IgG, indicating that the source of these glycans primarily is not contaminating blood (Figure 4B). The 441-27.2 vimin did not coelute with any standard and exhibited the *m*/*z* 407 fragment indicative of a GalNAc residue. The negCID D-18 ion at *m*/*z* 467, the F ion at *m*/*z* 465 and the core fucose specific *m*/*z* 350 fragment identified this glycan as M^3^AnF^6^ (Appendix A).

The 451 mass level harbored aside of A^4^GnF^6^bi and GnA^4^F^6^bi peaks yielded the *m*/*z* 407 indicative of LacdiNAc units. The 451-34.8 peak coeluted with the standard AnA^4^F^6^ and could be confirmed by neg CID (Appendix A). The 451-33.9 peak did not coelute with any standard and could not be identified by MS/MS.

The 452-32.9 was the only glycan of that composition. From the now already well known negCID fragments (Appendix A), it was identified as (AF)AnF^6^, which is in line with the reduced retention time compared to the only core fucosylated glycan A^4^AnF^6^ (36.9 vimin). Bovine kidney fucosidase treatment added this peak to the 451-23.3 peak, which by all tokens was identified as (AF)An.

The intense 541-22.7 representing A^4^A^4^F^6^ may in part be derived from residual blood. The compound 541-22.7, however, coeluted with Man4GnF^6^A-bi, which contains a bisecting lactose. Appearing in the desialylated fractions indicates previous sialylation, which must have been located on the bisecting lactose. Bisecting sialyl-lactose has not been described before to the authors knowledge.

Further structures of the 451 and 452 mass levels were (AF)A^4^, but not its arm isomer, (AF)A^4^F^6^ and A^4^(AF)F^6^ and a few unidentifiable peaks (Appendix A).

The mass levels containing five hexoses and five HexNAcs are shown in Figure 4E. The 550-15.1 was identified as A^4^A^4^bi, 551-22.3 as A^4^A^4^F^6^bi. The 552-19.9 peak contains the non-separable isomers A^4^(AF)F^6^bi and (AF)A^4^F^6^bi in an unknown ratio. A series of later eluting peaks (30.4, 32.5, 33.3, 35.0, 37.0 and 37.4 vimin) did not coelute with our standards and did not contain LacdiNac antennae, and are thus most probably triantennary glycans without a bisecting GlcNAc.

### 3.7. HNK-1 Structures

The desialylated sialo fraction contained six peaks giving posCID fragments *m*/*z* 542 and 622 identifiable as glucuronic acid (GlcA)+Gal+GlcNAc and SO4+GlcA+Gal+GlcNAc. Thus, these glycans contained the human natural killer cell determinant HNK-1 [49]. Based on this study, the largest peak was tentatively assigned the structure shown in Figure 5 based on literature data and maybe annotated as su^3^-Ga^3^-A^4^GnF^6^bi or more elegant NkGnF^6^bi. All HNK-1 containing structures were core-fucosylated. The extreme resistance towards negCID prevented further structural characterization. HNK-1 451a, HNK-1 451b and HNK 461 eluted at 28.1, 30.1 and 40.0 vimin, respectively. All other HNK-1 glycans eluted considerably after the last currently established time grid standard.

### 3.8. Occurrence of β1,3-Linked Galactose

Zamze et al. [11] found β1,3-linked galactose on biantennary glycans in the charged fraction of rat brain, whereas others did not find such glycans in the neutral fraction [9]. Β1,3-linked galactose is, however, rather common in the triantennary N-glycans, as reported by [18]. In the desialylated pool of this study, two peaks emerged that—according to their vimin values—harbored structures with a type I chain. The first peak (441-35.1) coeluted with A^3^GnF^6^ and was completely converted to GnGnF^6^ (341-28.1) upon galactosidase digestion. The second peak (542-35.5) coeluted with (AF)A^3^F^6^/A^3^(AF)F^6^ and was completely converted to the likewise inseparable isomer pair Gn(AF)F^6^/(AF)GnF^6^ (442-29.6) (Appendix A).

## 4. Discussion

Exact structures of 47 neutral and 21 desialylated N-glycans from the human brain were determined based on the application of biosynthesized reference glycans in standardized PGC-chromatography hyphenated to MS with negative and positive mode CID. With the exception of the just recently discovered glycans with substituted bisecting GlcNAc [28], many of the found structures have already been suggested by other studies, where methodological limitations often impeded the localization of the arm positions of terminal galactose and/or fucose residues [6,7,8,9,11,40,50]. Two large comparative studies of human [8] and mouse [7] brain N-glycomes used positive mode MS/MS for structural analysis. The rearrangements of certain sugar residues occurring during fragmentation in positive mode [23,25,28] and the very low ability of this method to discriminate arm-isomers constrained the authors to report compositions rather than explicit structures. The same limitations apply to approaches for de novo sequencing of N-glycopeptides using positive mode MS/MS [26].

Here, we identified 10 hybrid type glycans, 35 complex type glycans and 7 oligomannosidic glycans in the neutral pool and 2 hybrid type glycans and 21 complex type glycans in the desialylated pool. Furthermore, we identified six glycans carrying the HNK-1 epitope. Another 26 N-glycans of decent abundance whose structure could not be definitely resolved are characterized by their retention time and CID fragments, thus allowing unambiguous addressing in future studies (Appendix A). These 71 compounds include all N-glycans occurring at a relative abundance of more than about 5% compared to GnGnF^6^bi, which is the most abundant brain N-glycan. A list of all definitely identified N-glycans is shown in Figure 6. A list of all 98 monitored N-glycan peaks and their standardized retention times, and, when available, diagnostic fragment ions is given (Appendix A). The essential dissimilarity of natively neutral and de-sialylated structures is exemplified by a 2D map (Figure 7). In the following, a few interesting details will be discussed.

Rigorous data mining can retrieve many more glycan-related compounds [8]. We nevertheless settled for the major signals only as we adhere to the view that a structure-oriented investigation of a tractable number of compounds may be more helpful in revealing the underlying situation, i.e., the relative contribution of particular glyco-enzymes. Up- or downregulation of transferases may in certain cases only affect single minor structures, but we presume that it usually gets imaged in the major structures representing approximately 90% of the total peak area.

An observation, which could not have been made without the (in part asymmetrically) isotope-labeled standards, was that some pairs of arm-isomers coeluted on PGC. The isomer pairs (AF)Gn/Gn(AF), (F^2–4^)Gn/Gn(F^2–4^), (AF)A^4^F^6^bi/A^4^(AF)F^6^bi, (AF)A^4^bi/A^4^(AF)bi and (AF)GnF^6^/Gn(AF)F^6^ are examples of this phenomenon. When encountering these structures in real samples, wrong assumptions about the occurrence of isomers may be drawn. Awareness of this pitfall will guide the operator to have a close look at the D-ions in negCID, allowing a rough estimate of the arm isomer ratio [38].

Half of the identified neutral N-glycans carried a bisecting GlcNAc, which is typical for brain N-glycans [8,9,40]. *N*-acetylglucosaminyltransferase III (Gn-TIII) is indeed most highly expressed in brain and kidney [15]. Bisecting GlcNAc has a significant impact on the tertiary structure of a glycan [51], and thus may modulate the function of the target protein. Notable in this context, levels of bisecting GlcNAc are upregulated in brains from Alzheimer’s disease patients [52]. In several brain glycans, the bisecting GlcNAc is substituted with galactose or sialic acids. Interestingly, in all of these hybrid-type glycans, galactose was linked to the bisecting GlcNAc and not to the GlcNAc on the α1,3-arm, as presumed by other studies. In biantennary complex type glycans carrying bisecting GlcNAc residues, galactose (and LeX) were mainly bound to the antennal GlcNAcs, but smaller later eluting peaks most probably contained bisecting galactose as found in IgG and possibly even bisecting LeX. So, we conclude that galactosylation of bisecting GlcNAc in biantennary glycans is slow but possible, whereas galactosylation of the three-arm GlcNAc in Man4-hybrid type glycans is strongly impeded.

It is commonly held that brain N-glycans are modified with LeX fucose [9,11,18,40,53]. Indeed, no brain glycans coeluting with the biosynthesized LeA standards were identified—neither in the neutral nor in the desialylated fraction. Notably, discrimination between Lewis epitopes is often evaded as e.g., in a large structural study of the human brain N-glycome [8]. Likewise, we could not identify glycans with α1,2-linked fucose. This is in line with some studies [8,9,11], while others found α1,2-linked by lectin or antibody binding or in vivo magnetic resonance spectroscopy [36,37,54]. Thus, the α1,2-linked fucose is either found on very minor N-glycans or on other scaffold structures.

We detected six glycans modified with the HNK-1 epitope. A glycan with the composition H4N5F1 + HNK-1 expectably has the structure found on a myelin glycopeptide [49]. We identified two isomers with this composition and another four HNK-1 containing peaks (Figure 6). Unfortunately, all HNK-1 containing glycans exhibited extreme resistance towards negative-mode CID, preventing branch allocation of the HNK-1 chain. Notably, the brain expresses the isozyme GlcAT-P, which in contrast to GlcAT-S is dampened by bisecting GlcNAc [55]. This interdependence emphasizes the potential relevance of bisecting GlcNAc regarding physiological functions.

Eight out of ten hybrid type glycans in the neutral pool were substituted with a bisecting GlcNAc, which in four glycans was even β1,4 galactosylated—resulting in bisecting lactose—or elaborated to a bisecting Lewis X epitope [28]. The structure with galactosylated GlcNAc (M^3^GnF^6^A-bi) is also present in the desialylated glycan pool, indicating a sialic acid residue on the “bisecting” galactose prior to desialylation. A glycan with this modification has not been described before in the brain N-glycome. Glycans containing sialylated “bisecting” galactose were previously identified in human serum IgG on biantennary rather than hybrid-type glycans [56].

An unusual H3N4F1-isomer substituted with a bisecting GlcNAc but not GlcNAc on the α1,3-arm (Appendix A, 14.7 vimin) can be explained as the product of a brain specific hexosaminidase B [43]. We detected three additional glycans lacking a terminal GlcNAc on the three-arm, which are probably attributable to this hexosaminidase B (Figure 4).

The structures of the neutral and desialylated pools exhibited little congruence. This disparity of neutral and desialylated pools raises the question as to the origin of the phenomenon. Are different cells or rather biosynthetic branching points with no return option responsible? The biosynthesis of LeX epitopes in the brain is predominantly catalyzed by fucosyltransferase 9 (Fuc-TIX), which is not able to biosynthesize the sialyl Lewis X (sLeX) determinant [57,58]. The neutral fraction in fact contains a number of glycans with LeX antennae. (Appendix A, Figure 2). On the contrary, all originally sialylated structures contained at least one antenna with a free galactose. Thus, we conclude that once an antenna has been fucosylated, it no longer serves as a substrate for the sialyltransferases, as there is no mammalian sialyltransferase acting on LeX epitopes [57]. All identified glycans in the desialylated fraction in our study contained at least one galactosylated antenna without antennary fucose, and we assume that this is the arm bearing the sialic acid.

Only recently has a LacdiNAc (GalNAc-GlcNAc-) containing structure been described to occur in the brain [8]. In organs other than the brain, glycans containing LacdiNAc-units confer important biological functions as in self-renewal of embryonic stem cells from mouse [59] or malignancy in certain types of tumors [60]. B4GalNAc-T4, the enzyme probably responsible for LacdiNAc formation in the brain, is highly expressed in the fetal and adult brain, pointing out the potential significance of this epitope in the brain [61]. We identified six GalNAc-containing glycans in the desialylated fraction and three in the neutral fraction. With the help of MS/MS, we could fully identify one hybrid-type glycan with a GalNAc residue-Man4AnF^6^ (probably, but not substantiated M^3^AnF^6^) in the desialylated glycan pool and one doubly fucosylated hybrid-type glycan with a GalNAc residue—Man4(AnF)F^6^—in the neutral fraction. This again underscores the inhibitory role of fucosylation for sialylation. Much more interestingly, this reveals the presence of fucosylated LacdiNAc units in a mammalian tissue, also known as LDNF. This structure is typical for some parasitic worms [62,63,64,65] although it has been, albeit rarely, found in mammalian proteins [45,66,67] and also in insect allergens [68].

## 5. Conclusions

The application of standardized PGC chromatography combined with MS/MS has led to the complete structural assignment of 68 human brain N-glycans. Application of internal standards confers coordinates (mass + time + characteristic CID fragments) for all peaks, including currently unidentifiable structures. This coordinate system allows to precisely trace all glycan isomers and refer to them in future studies. With the expectable addition of reference glycans for sialylated N-glycans, the time-grid based approach may form the basis of rational deep structural glycomics that can build and rely upon previous structural assignments, rather than starting from scratch over and over again. Clearly, the approach is applicable to tissues other than brain.

## Figures and Tables

**Figure 1 biomolecules-12-00085-f001:**
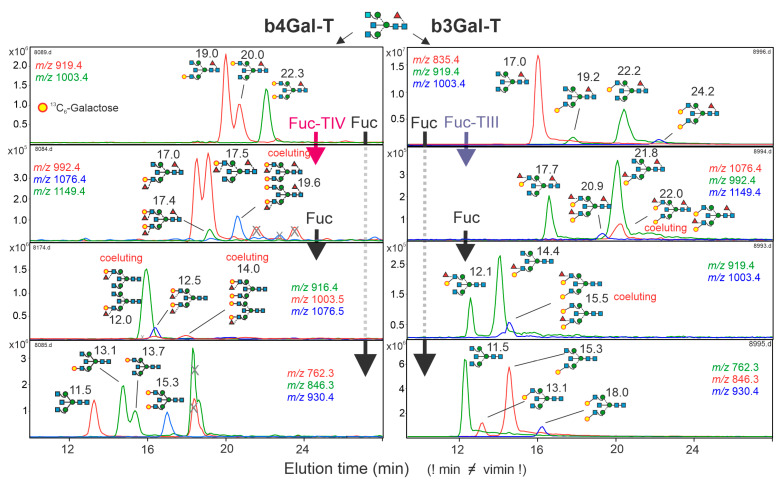
Biosynthesis of biantennary glycan standards with bisecting GlcNAc containing one or two galactose and zero to three fucose residues. The left and right side show the PGC-ESI-MS chromatograms of glycans containing β1,4-galactose or β1,3-galactose, respectively. GnGnF^6^bi was treated with either b4Gal-T and b3Gal-T5 using ^13^C_6_-UDP-galactose as the donor substrate. The products were subjected to different sequences of Lewis-type fucose incorporation and core-fucose removal by either Fuc-TIII or Fuc-TIV or bovine kidney fucosidase (Fuc). Retention times were normalized with the help of the Tigr-Mix [28], hence the discrepancy between measured and virtual retention times (“vimin”), as given above the structure cartoons. Peaks with an incongruent isotope pattern are marked by an X.

**Figure 2 biomolecules-12-00085-f002:**
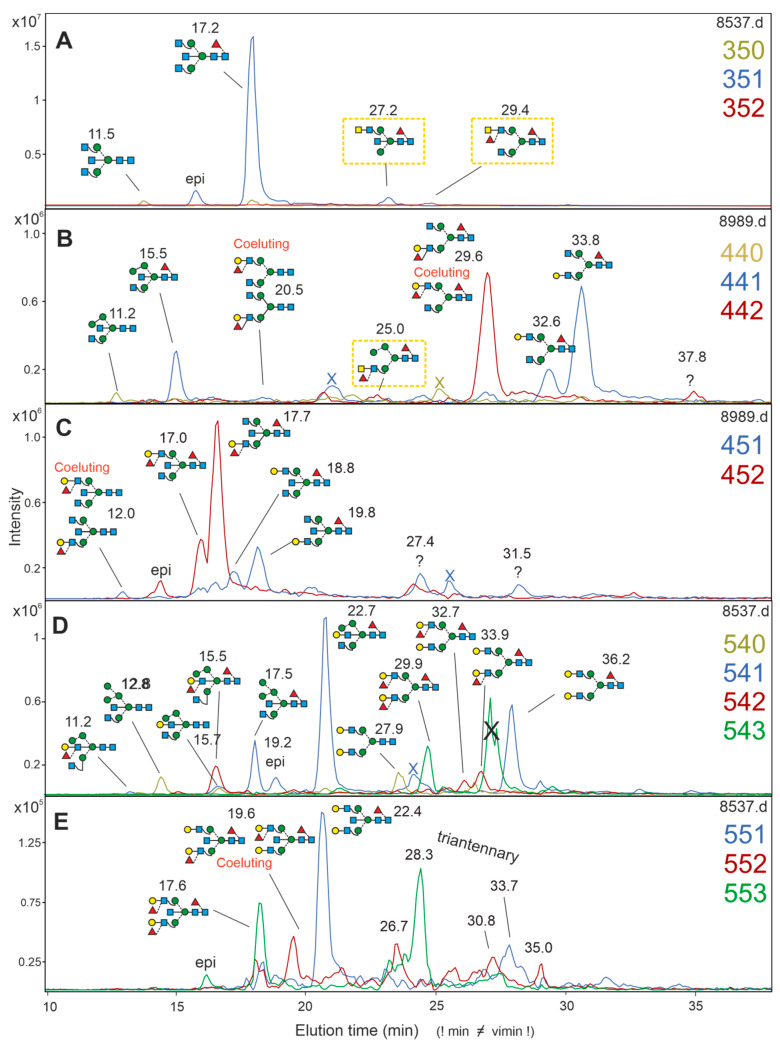
Analysis of neutral N-glycans from the diencephalon by PGC-LC-MS. Extracted ion chromatograms for [M+2H]^2+^ ions are shown for glycans of the selected compositions, as indicated on each panel’s right side by the number of hexoses, HexNAcs and fucoses. Yellow boxes highlight GalNAc containing glycans. Structures for which no reference glycans were available are marked by ecru background, tentative assignments by pink background. Peaks with vimin labels only constitute N-glycans of yet undefined structure. Mass values and CID details for all peaks are found in the comprehensive brain N-glycan repertoire (Appendix A). Panel (**A**–**E**) show the EICs for the glycan compositions 350, 351, and 352; 440, 441, and 442; 451 and 452; 540, 541, 542, and 543; and 551, 552, and 553, respectively as indicated by at the right side of each panel by numbers in matching colors.

**Figure 3 biomolecules-12-00085-f003:**
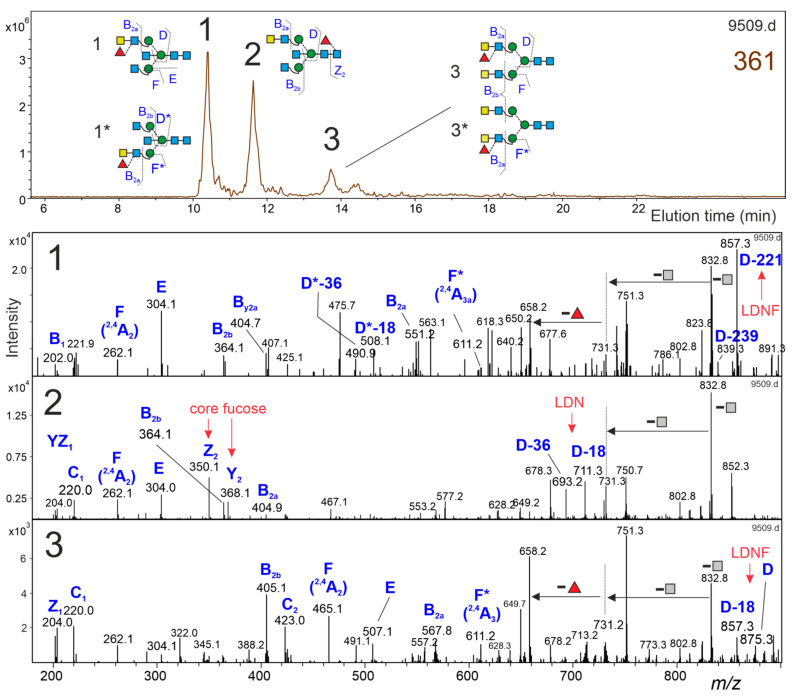
Glycans with fucosylated LacdiNac (LDNF) antennae. A HILIC fraction of an N-glycan preparation from the Lobus frontalis was treated with bovine kidney fucosidase. The extracted ion chromatogram of the 361 mass level in an accelerated PGC-LC-ESI-MSMS run exhibited three peaks. Two peaks with D-ions indicative of LDNF elements could be clearly distinguished from a residual core-fucosylated glycan by their negCID spectra. The structures marked with an asterisk show the possible arm-isomers represented by the respective peak. The fragments marked with an asterisk in the MS/MS spectra indicate fragment-ions derived from the possible arm-isomer.

**Figure 4 biomolecules-12-00085-f004:**
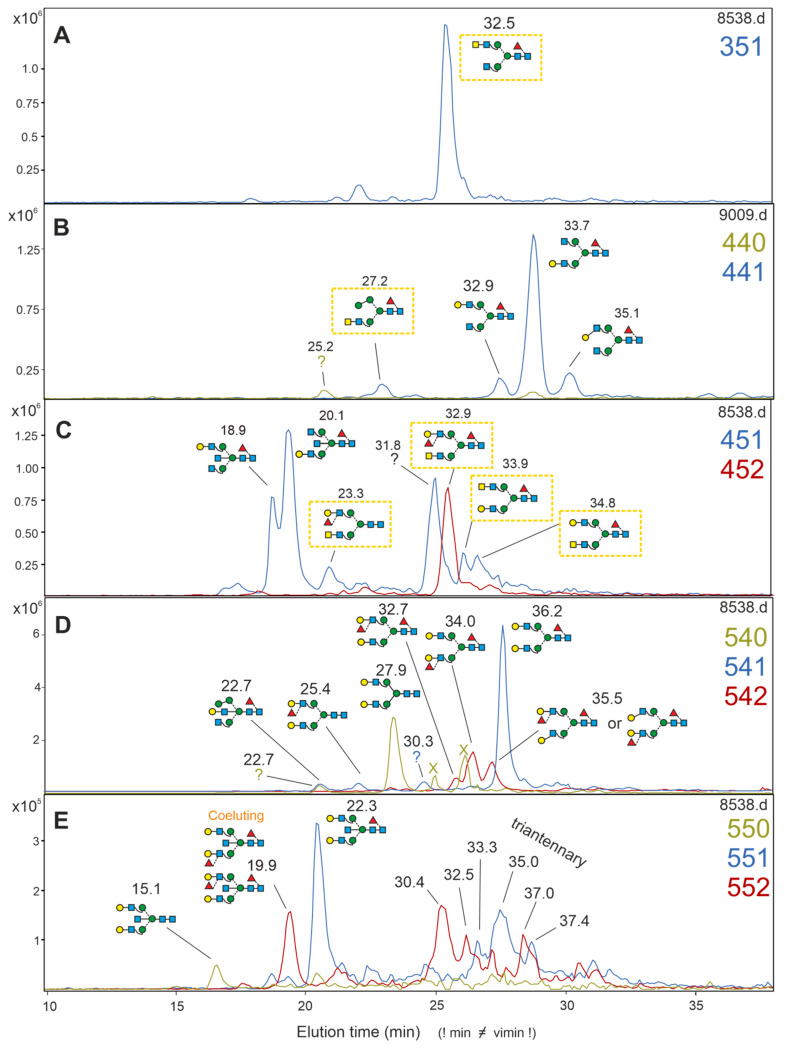
Analysis of the previously acidic, desialylated N-glycans from the diencephalon by PGC-LC-MS. Extracted ion chromatograms for [M+2H]^2+^ ions are shown for glycans of the selected compositions, as indicated on each panel’s right side by the number of hexoses, HexNAcs and fucoses. Yellow boxes highlight GalNAc containing glycans. Vimin numbers are given above each structure or unidentified peak, while the *x*-axis shows the retention time of the particular experiment. Peaks with an incongruent isotope pattern are marked by an X. Mass values and CID details for all peaks are found in the comprehensive brain N-glycan repertoire (Appendix A). Panels (**A**–**E**) show the EIC for the glycan compositions 351; 440 and 441; 451 and 452; 540, 541, and 542; and 550, 551, and 55, respectively, as also indicated by the numbers in matching colors.

**Figure 5 biomolecules-12-00085-f005:**
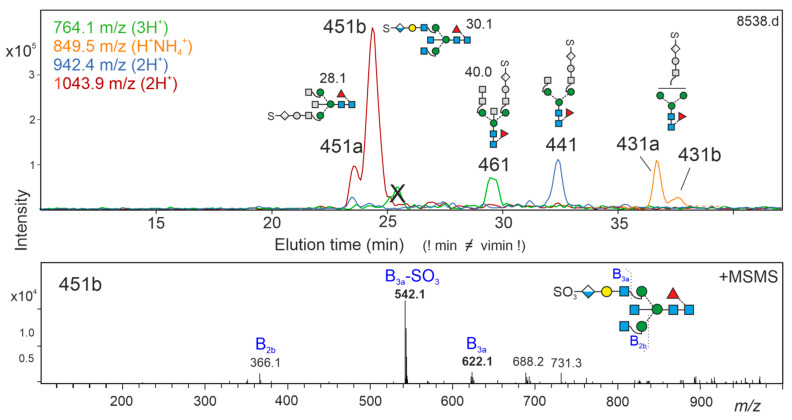
HNK-1 containing N-glycans. Extracted ion chromatograms for peaks yielding posCID spectra with *m*/*z* = 542.1 and 622.1 (sulfate_0-1_-glucuronic acid-galactose-GlcNAc) are shown together with the MS/MS spectrum of the major peak, which is bona fide assumed to represent the HNK1-containing structure previously found in nerval tissues and elucidated by NMR [39,49]. Structure cartoons partly drawn in grey are tentative explanations of the respective composition.

**Figure 6 biomolecules-12-00085-f006:**
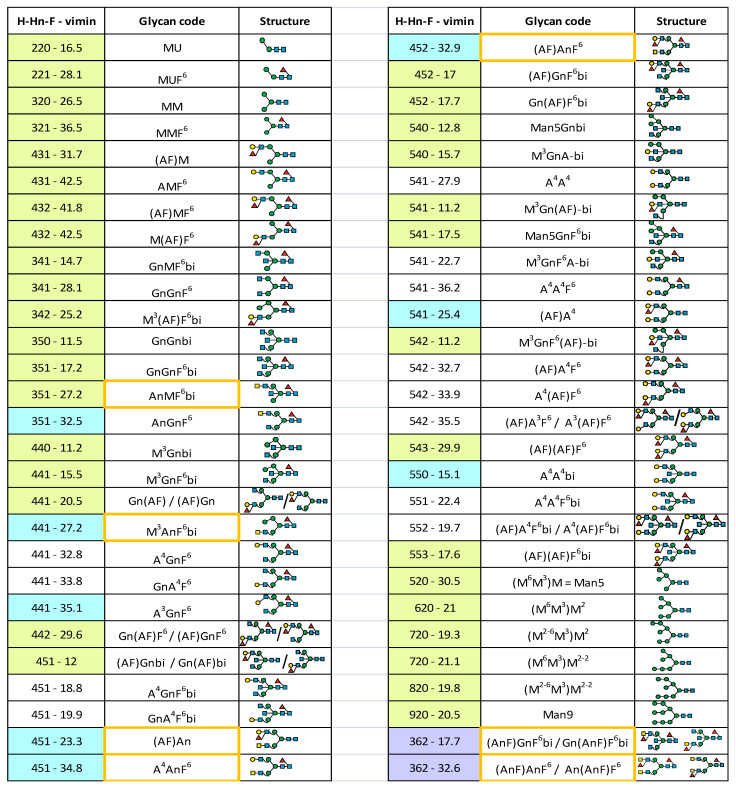
Structures of the major brain N-glycans. Compositions are given by the number of hexoses, *N*-acetylhexosamines and fucose residues together with their virtual retention times, which shall be read with a ±0.1 min error tolerance. HNK-1 containing glycans as well as other sulfated or phosphorylated N-glycans that occur in the brain [8,39] are not considered in this table. Glycans with GalNAc are highlighted by amber borders. The listed structures were found in the diencephalon except for the LDNF-containing glycans, which emerged in the *Lobus frontalis*. Regarding glycan codes, a detailed explanation of this abbreviation system can be found in the Supporting Information of [28].

**Figure 7 biomolecules-12-00085-f007:**
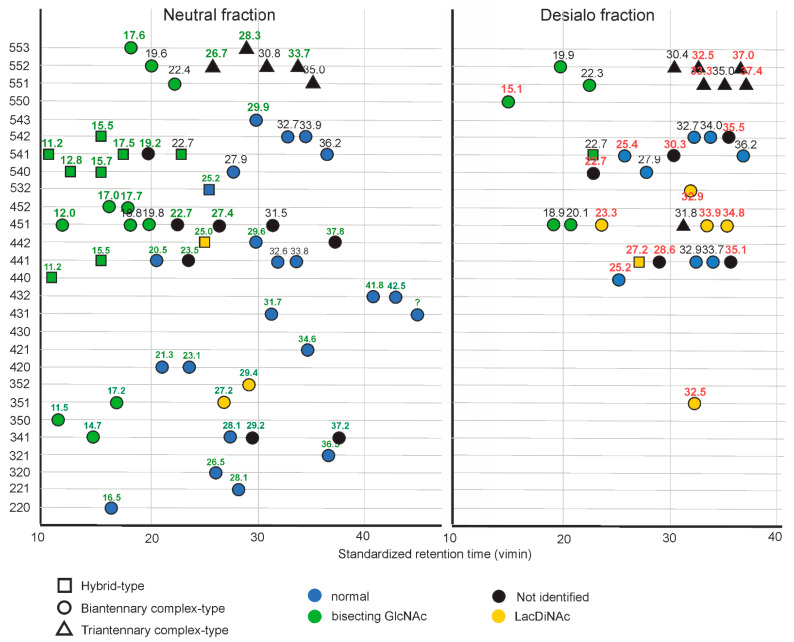
Plot of virtual retention time versus composition of N-glycans from the human diencephalon. Originally neutral and desialylated structures are plotted separately. Vimin labels for structures occurring in both fractions are written in black.

## Data Availability

The data presented in this study are available in the Appendix A or upon request from F.A.

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
