# Peer review of "Towards Mapping of the Human Brain N-Glycome with Standardized Graphitic Carbon Chromatography"

_biomolecules, 2022, doi:10.3390/biom12010085_

Round 1
Reviewer 1 Report
In this study, Helm J. et al. describe a novel method using standardized PGC chromatography combined with positive and negative CID-MS/MS for isomer-specific analysis of human brain N-glycome. By modifying standard glycan structures with glycosyltransferases and glycosidases, the authors biosynthesized a variety of N-glycan standards. The repertoire includes N-glycans with typically observed features in the N-glycome and thus was optimized for the comprehensive analysis of this tissue. For the LC-based comparison of endogenous glycan structures with the reference standards, the authors employed virtual retention time (vimin) based on the retention time of internal standards. By comparing the neutral and desialylated fractions, sialylated glycans and corresponding moieties could be identified. The main finding of the analysis of human brain N-glycans is the identification of minor but obvious structures including NNK-1 and LNDF epitopes as well as bisecting sialyl-lactose.
This manuscript provided a valid strategy to perform isomer-specific analysis of endogenous N-glycome, which is an important limitation of current glycomics technology. The authors clearly described the detailed materials and methods, as well as strengths and weakness (limitations) of the present method, citing appropriate references.
However, the reviewer think that this manuscript requires many revisions to be considered for publication, as listed below.
Major comments:
In Table S18, the HPLC chromatograms showed an obvious shift of peaks corresponding to neutral glycans before and after sialidase digestion. This time lag seems to be larger than expected with the same condition. Please explain this reason.
Minor comments:
Line 13: “N” of N-glycan and N-glycome should be italicized.
Line 42: It would be better to indicate representative references of large structural and functional studies regarding the brain N-glycome.
Line 107: Please add explanations about the abbreviations of glycan structures such as A4A4. It may also be better to note that the glycan structure list is indicated in Table S1.
Line 123: In my opinion, “b3GalT” seems unusual and “B3GalT” or “β3GalT” may be better for the abbreviation of β1,3-galactosidase. In addition, please use the same abbreviation throughout the manuscript for the clarity. For example, “b1,3-GalT” and “β1,3-GalT” are also used as the abbreviations in Figure 1.
Figure 1: This figure may be confusing regarding the retention time. This Figure caption seems to be incomplete. It may be better to simply indicate as “Elution time (min)” in the horizontal axis and add the descriptions for “vimin” indicated within the figures. It also may be better to explain “vimin” in the main text (Line 218), not only in the Figure caption.
Line 201: Please spell out “bgH”.
Line 209: SI Table 1 should be replaced by Table S1.
Line 227: It may be better to move this paragraph to the Discussion section.
Line 250: The correct one may be not Fig. 2D map but Figure 7.
Line 263: The correct one may be not 350-11.7 but 350-11.5, based on Table S1.
Line 264: “LacDiNAc” should be replaced by “LacdiNAc”.
Line 270: Could you explain more clearly the meaning of the sentence “Hence, ~.” In addition, the correct one may be not An4GnF6 but A4GnF6, based on Table S1.
Line 288: The correct one may be not 451-11.2 but 451-12.0, based on Table S1.
Line 302: It may be better to move this paragraph to the Discussion section.
Line 309: Figure 4 appears not to contain the results regarding this sentence.
Line 375: Figure 4B appears not to contain the results regarding this sentence.
Line 389: The correct one may be not 362-32.4 but 451-32.9, based on Table S1. Please check Line 392 together.
Line 405: No description regarding Figure 5A is indicated.
Line 408: As you utilized the abbreviations shown in Table S1 throughout the manuscript, “Man4AnF6“ may be replaced by “M3AnF6“ for the clarity.
Line 415: The correct one may be not A4AnF6 but AnA4F6, based on Table S1.
Line 416: Please explain (spell out) BK fucosidase.
Figure 3: The dotted line showing the glycan fragmentation is hard to see.
Line 432: The authors described that mass levels of 550, 551, and 552 were “rather populated”, but the vertical axis is “×105” in Figure 4E, implying the lower abundance of these glycan structures compared to that of the others shown in Figure 4.
Line 433: The correct one may be not 552-20.2 but 552-19.9, based on Table S1.
Line 435: Only three of the indicated peaks are shown in Table S1.
Line 443: The correct one may be not Figure 6 but Figure 5.
Line 449: Please check the number of this subsection title.
Figure 6: The figure is missing! In the Figure caption, the correct one may be not FDNL but LDNF.
Line 560: The correct one may be not 14.7 vimin but 18.9 vimin, based on Figure 4C.
Author Response
Please notice the attached file.

Reviewer 2 Report
After having read the manuscript entitled "Towards Mapping of the Human Brain N-Glycome with Standardized Graphitic Carbon Chromatography", I came to the conclusion that the manuscript has a high quality to be published in Biomolecules. Minor revision should be made before publication.
- The abbreviations such as GnGnF6 , A4GnF6 , GnA4F6 and A4A4F6 should be explained clearly in the main text.
- In Line 163, H20 should be change to H2O.
- In Line 199-200, 13C2, A4A4 and 13C6 should be revised to superscript or subscript.
Author Response
Reviewer 2:
We thank the reviewer for this positive perception of our work.
After having read the manuscript entitled "Towards Mapping of the Human Brain N-Glycome with Standardized Graphitic Carbon Chromatography", I came to the conclusion that the manuscript has a high quality to be published in Biomolecules. Minor revision should be made before publication.
- The abbreviations such as GnGnF6 , A4GnF6 , GnA4F6 and A4A4F6 should be explained clearly in the main text.
Answer: Already in response to Reviewer 1, references to Tables and explanation were added.
- In Line 163, H20 should be change to H2O.
- In Line 199-200, 13C2, A4A4 and 13C6 should be revised to superscript or subscript.
Answer: We thank the reviewer for these hints